# Influence of the Cooling Method on Cutting Force and Recurrence Analysis in Polymer Composite Milling

**DOI:** 10.3390/ma17235981

**Published:** 2024-12-06

**Authors:** Krzysztof Ciecieląg

**Affiliations:** Department of Production Engineering, Faculty of Mechanical Engineering, Lublin University of Technology, 36 Nadbystrzycka, 20-618 Lublin, Poland; k.ciecielag@pollub.pl

**Keywords:** polymer composites, milling, cutting force, cooling, recurrence method, recurrence quantifications, ANOVA analysis

## Abstract

This work investigates the milling of the surface of glass and carbon fiber-reinforced plastics using tools with a polycrystalline diamond insert. The milling process was conducted under three different conditions, namely without the use of a cooling liquid, with oil mist cooling, and with emulsion cooling. The milling process of composites was conducted with variable technological parameters. The variable milling parameters were feed per tooth and cutting speed. The novelty of this work is the use of recurrence methods based on the cutting force signal to analyze the milling of composites with three types of cooling. The primary aim of the study was to determine the effect of variable technological milling parameters on cutting force and to select recurrence quantifications that would be sensitive to the cooling method. It has been shown that recurrence quantifications such as determinism (DET), laminarity (LAM), averaged diagonal length (L), trapping time (TT), recurrence time of the second type (T_2_), and entropy (ENTR) are sensitive to the cooling methods applied for the tested composite materials. The results have shown that it is possible to determine common ranges of changes in sensitive recurrence quantifications for the two tested variables parameters of milling: 0.63–0.94 (DET), 0.69–0.97 (LAM), 7.30–13.48 (L), 2.92–4.98 (TT), 17.01–38.25 (T_2_), 2.02–3.16 (ENTR). The ANOVA analysis results have confirmed that the studied variables have a significant impact on the recurrence quantifications.

## 1. Introduction

Polymer composites are widely used in industry due to their specific properties. The properties that distinguish composites from other engineering materials include their high tensile strength at low specific weight [1]. In addition, polymer composites are characterized by their ability to dampen vibrations and ease of forming any shape, high corrosion resistance, as well as good electrical insulating properties. The properties of a composite consisting of reinforcement, matrix, and additives are not the sum or average value of the properties of its individual components, but rather a new material is obtained, exhibiting higher properties than each of its components separately [2]. The parameters of a given composite depend on the properties of matrix and reinforcement, as well as on the relative content of fibers and matrix calculated as volume fractions [3].

Composites can be manufactured using many methods, each of them associated with labor and costs. The process of obtaining a composite finished product is associated with machining to achieve the required roughness, shapes, and dimensional tolerances [4]. Composite machining differs significantly from operations performed on metals and their alloys due to the anisotropy and inhomogeneity of the composite structure. In addition, composite materials are difficult to machine, with high temperatures generated affecting their dimensional accuracy [5]. Therefore, dedicated tools are used for composite machining [6]. In addition to that, machining processes for composites are conducted with cooling fluids. They are widely used in the processing of materials such as titanium alloys or steel [7]. In metal machining, the standard practice is to use an oil-based coolant or an oil-water emulsion. The purpose of the coolant is to remove heat from the tool and the element, as well as to provide lubrication, allowing for higher cutting speeds while reducing the risk of damage to the workpiece and the tool.

A review of the literature demonstrates that polymer composites, especially those reinforced with carbon fibers (CFRP), are sensitive to moisture. Moisture has a harmful effect on composites, therefore the method of using liquid and its amount should be appropriately selected [8]. The effect of water absorption on adhesive joints inside the composite and mechanical properties was studied. It was shown that water absorption would deteriorate mechanical properties such as fatigue, impact resistance, or bending strength, which are primarily related to the properties of the matrix [9]. In contrast, contact with moisture does not affect tensile strength. After water desorption, the effects of the liquid are irreversible. Irreversible effects can be caused by long-term or cyclical use of various types of liquids. It was shown that mechanical properties decreased with the amount of moisture absorption [9]. The effectiveness of coolant use was also tested in terms of its effect on machinability parameters such as cutting forces, torque, surface roughness, thermal wear of the machined surface, and tool wear. It was found that the use of coolant in machining polymer composites reinforced with carbon fibers led in most cases to reduced torque, cutting forces, and tool wear, as well as to improved surface roughness [10]. A study also showed that machining coolants caused less degradation of mechanical properties than water [8]. Another advantage of using coolants is the reduction in the extent of composite surface damage caused by heat [11]. A smaller amount of heat, through lower values of the resultant cutting force and tool wear, has a positive effect on surface quality. In order to achieve a compromise between the advantages and disadvantages of using coolants, popular cooling media such as minimal lubrication (MQL), cryogenic-liquid nitrogen (N2_liquid_), and carbon dioxide (CO2_ice_) were tested and shown to reduce the resultant cutting force, tool wear, surface roughness and cutting temperature [11]. In terms of short-term impact on the surface of polymer composites, their effect on the surface depends on the material components. These cooling agents are known as sustainable cutting fluids. They are an alternative to cutting fluids used in metalworking because they are more environmentally friendly. In addition, they provide lower temperature, cutting force, lower vibrations, lower delamination percentage, and extended tool life. Their physicochemical properties offer wide possibilities for their application in composite machining [12,13]. The use of cooling–lubricating fluid and cold air as a coolant in CFRP machining by the rotational ultrasonic method mainly leads to reduced torque and cutting forces, as well as improved surface roughness, reduced tool wear, and elimination of burnout on the machined surface [8]. The results of experimental studies also showed that sustainable cooling could also improve tribological conditions by reducing cutting temperatures, cutting energy as well as wear of the side surface. The MQL-based lubrication strategy provides the best tool wear index and surface characteristics, i.e., surface roughness and surface topography [14]. The effectiveness of using sustainable cutting fluids techniques was confirmed by the Taguchi method and analysis of variance (ANOVA). The influence of cutting parameters on surface roughness, flank wear, and cutting temperature was evaluated using cooling techniques [15]. For hole-making by drilling, it was found that drilling CFRP with coolant yielded better surface quality than dry drilling [16]. Experimental studies comparing cutting force, torque, surface roughness, burn of machined surface, and tool wear in machining with different types of coolants also showed the beneficial effects of coolants on machinability indices [10].

Polymer composite machining is also exposed to damage, including delamination, fiber pullout, and matrix cracking [17]. Damage affects the efficiency of machining and can lead to disqualification of the machined element. In addition, the matrix serving as a binder connecting the fibers has low heat resistance, leading to thermal damage and degradation of the composite structure. To prevent moisture absorption into the composite structure and to affect the dimensional stability and physical properties of a composite element, sustainable cutting fluids are used, including minimal lubrication (MQL), cryogenic-liquid nitrogen (N2_liquid_), and carbon dioxide (CO2_ice_) [18]. It was also observed that balanced cutting fluids such as cryogenic-liquid nitrogen (N2_liquid_) and carbon dioxide (CO2_ice_) introduced into the machining zone during cutting titanium alloys would reduce tool wear, temperature, and cutting force [13]. In contrast to traditional cooling, sustainable cutting fluids effectively reduce thermal wear and cutting temperature. Studies on the effect of cryogenic-liquid nitrogen (N2_liquid_) on the machinability parameters in cutting titanium alloys showed a 66% decrease in cutting temperature, 36% decrease in surface roughness, 42% decrease in cutting power and 39% decrease in flank wear compared to traditional lubrication. Reduced cutting temperature affects tool wear and, consequently, causes a 39% reduction in surface roughness [19]. The use of sustainable cutting fluids also causes a six-fold increase in tool life by reducing the amount of generated heat [20,21]. Due to environmental issues, liquid nitrogen is often used as a coolant, because it returns to gaseous form by evaporation without harming the environment [22,23]. In machining what is also important to use is feed rate control, especially in the case of curved toolpaths, where the angular velocity of movement is involved, which happens also in standard applications of composite milling [24,25].

Machining is associated with cutting forces [26,27], which can be used for process analysis. In the case of milling and drilling, one of the components of the total cutting force, i.e., the cutting force can be used for process analysis [28]. This force can be used for cutting process analysis by recurrence methods [29]. These are methods based on the analysis of nonlinear signals [30]. The first stage of such analysis is to reconstruct the delay vector *x* [31]. Reconstruction involves the selection of the time delay *d*, the embedding dimension *m*, and the threshold ε [32]. The reconstructed delay vector *x* is described by Formula (1):*x* = (x_i_, x_i+d_, x_i+2d_, …, x_i+(m−1)d_)(1)

Based on selected parameters, a recurrence plot is also created, consisting of individual points and a texture created by vertical and horizontal lines. Periodic signals on the plot are usually presented using diagonal lines or repeating symmetrical structures. Recurrence diagrams can only provide qualitative information, which is why recurrence quantifications were introduced [33,34,35]. In combination with recurrence plots, they are an advanced tool for analyzing nonlinear signals.

The effective use of recurrence analysis allowed its application in machining. Due to the fact that machining is a nonlinear dynamic process, researchers have studied the application of this method in analyses of processes such as milling [36,37], drilling [38], or turning [39]. In addition to turning control using recurrence methods, research is also conducted on the assessment of surface roughness based on acoustic signal emission [40]. In the field of turning, it was also found that recurrence analysis is a suitable method for determining the point of transition from vibration-free machining to machining with vibrations [41]. In drilling, recurrence methods were used to monitor the condition of the cutting tool in order to obtain the appropriate hole quality in drilling carbon fiber-reinforced plastics [38]. Recurrence methods offer great potential in the study of nonlinear systems in composite drilling to determine cutting parameters and hole diameter depending on vibrations [42]. In the case of drilling, recurrence quantifications also allow for detecting the size and location of defects in the machining of glass and carbon fiber-reinforced plastics [43]. A widely developing area of machining with numerous applications is milling, in which recurrence methods have also been applied. The use of this nonlinear analysis made it possible to determine the approximate boundary between stable vibration-free and unstable machining by milling [44]. A study showed that recurrence methods made it possible to make the values of recurrence quantifications dependent on variable cutting speed during milling [45]. In addition, recurrence quantifications such as laminarity (LAM), trapping time (TT), and entropy (ENTR) were determined, enabling the detection of the wear of cutting inserts based on the vibration signal [36]. Conducting milling with a simultaneous analysis of this process by recurrence methods allowed the determination of recurrence quantifications for defect detection [29].

The literature review has shown that the use of recurrence quantifications makes it possible to study and determine phenomena occurring during the machining process. Previous research on sustainable cooling in composite machining has demonstrated that the use of MQL technology has a beneficial effect on machinability indicators such as cutting forces. This study investigates two types of polymer composites reinforced with glass and carbon fibers, supersaturated with epoxy resin, which are subjected to milling under dry milling conditions as well as using MQL technology and emulsion cooling. A novelty of this study is that it also investigates cutting force and its use for analysis by recurrence methods. The aim of the study is to determine recurrence quantifications that are sensitive to the type of coolant and material.

## 2. Materials and Methods

Milling operations were performed on a vertical machining center, Avia-VMC 800 HS. Experiments were conducted using Seco tools. Milling was performed using a 12 mm diameter cutter consisting of a body (symbol R217.69-1212.0-06-2AN) with two polycrystalline diamond (PCD) inserts (symbol XOEX060204FR PCD05). The tests were conducted with variable feed per tooth ranging from 0.025 to 0.3 mm/tooth and variable cutting speed in the range of 50–600 m/min. The milling operations were performed at a constant cutting depth of 1 mm.

The research was conducted for three cooling variants. The first one consisted of “dry” milling, i.e., without the use of a cooling liquid. The second one involved using the MQL technology where the factor was Mobil VG68 oil. In the third approach, cooling was realized using the Mobilcut 230 emulsion which was introduced via nozzles making part of the equipment of the vertical machining center.

Test samples were two composite materials: glass fiber-reinforced plastic (GFRP) with the trade name EGL/EL 3200-120 and carbon fiber-reinforced plastic (CFRP) with the trade name HexPly AG193PW/3501/6SRC41. Both materials are saturated with epoxy resin. The composite samples were in the form of plates with the dimensions of 10 × 100 × 400 (Figure 1). The 10 mm thick sample was formed by layers of 40 prepregs in a 0°–90° arrangement. The composite samples were fabricated in an autoclave where they were heated for 2 h at 177 °C (+/−2 °C) at a pressure of 0.3 MPa. Prior to putting them into the autoclave, the samples were prepared in a special room where cleanliness was maintained and the number of solid particles per 1 m^3^ did not exceed 10,000. The temperature in the room was maintained in the range of 18 °C–30 °C and the humidity level was below 60%.

A scheme of the experimental setup is shown in Figure 1.

Force values were measured on the machining center using Kistler’s 3D dynamometer (type 9257B) connected to a Kistler charge amplifier (type 5070). Signals were recorded with a Dynoware data acquisition card (type 5697A) and DynoWare V2.6.3.12 software from 30 September 2013 (type 2825A). The calculations of recurrence quantifications were performed using Matlab 2022 software. Table 1 presents recurrence quantifications and their formulas.

The table in the formulas contains symbols that are needed to calculate the recurrence quantifications: *i* and *j* are the numbers of states in space, *N* is the number of points analyzed in phase space, *x_i_* is the *i*-th vector and *x_j_* is the *j*-th vector, *l* is the length of the diagonal line, *P*(*l*) is the histogram of the length of the diagonal lines, *v* is length of vertical line, *P*(*v*) is histogram of the length of vertical lines, *N_l_* is the number of diagonal lines, *N_v_* is the number of vertical lines, *H_v_*(*v*) is the distribution of vertical lines, *p*(*l*) is the probability of diagonal-line length distribution, *k* is number of neighbors, *R*_*j*,*k*_ is the number of neighbors around *j*-th point, *R*_*k*,*i*_ is the number of neighbors around *i*-th point.

## 3. Results

The milling process was conducted using three different cooling media and variable technological parameters: feed per tooth and cutting speed. The milling process was conducted with a constant cutting depth of 1 mm. The subsections below describe the effect of variable feed per tooth on the recurrence quantifications when milling two different composite materials by three different cooling methods, i.e., using “dry” machining, MQL technology, and emulsion. The signal for analysis was the measured cutting force.

### 3.1. Effect of Technological Parameters on Cutting Force

In order to determine the influence of variable technological parameters on recurrence quantifications, the maximum cutting force was estimated. Figure 2a,b show the influence of variable feed per tooth and cutting speed on the maximum cutting force value. This force value was then subjected to recurrence analysis.

An analysis of the plots reveals that the maximum cutting force increases for both materials with an increase in the feed per tooth and cutting speed. The highest values were achieved for the “dry” machining of both materials. However, the maximum values of the cutting force were higher for CFRP than for GFRP. The highest maximum cutting force of 248 N was achieved in the milling process of carbon fiber-reinforced plastic conducted with a feed per tooth of 0.3 mm/tooth and without the use of coolants. The lowest value of the maximum cutting force of 60 N was achieved in the milling process of glass fiber-reinforced plastic conducted with a cutting speed of 50 m/min using MQL technology.

### 3.2. Effect of Feed per Tooth on Recurrence Quantifications

The first of the analyzed milling parameters was feed per tooth to determine its effect on recurrence quantifications in milling glass fiber-reinforced plastic (GFRP) and carbon fiber-reinforced plastic (CFRP) using dry machining (dry), MQL technology (MQL) and emulsion fed through the machine nozzles (emulsion).

Diagrams in Figure 3, Figure 4, Figure 5, Figure 6, Figure 7 and Figure 8 show the relationship between recurrence quantifications and feed for the two tested materials and different cooling media.

An increase in the feed per tooth value causes an increase in determinism (Figure 3a) from 0.63 to 1.02. The highest values of determinism (DET) for the feed per tooth (0.3 mm/tooth) were achieved during the dry machining of GFRP. The lowest determinism values were achieved in CFRP machining using emulsion for lower feed per tooth values (0.025 mm/tooth). In the laminarity diagram (Figure 3b) there is a noticeable decrease (from 0.97 to 0.69) in the value of this quantification with increasing the feed per tooth. The highest laminarity (LAM) value was obtained for GFRP machining using emulsion and the lowest feed per tooth. The lowest laminarity value was achieved in the milling of CFRP material that was conducted using MQL and a feed per tooth of 0.3 mm/tooth.

**Figure 3 materials-17-05981-f003:**
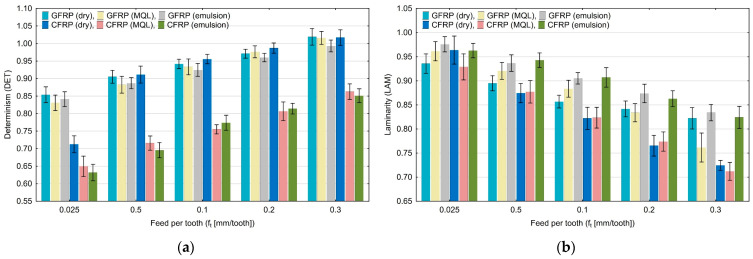
Influence of feed per tooth on recurrence quantifications: determinism (**a**), laminarity (**b**).

Figure 4a shows that with increasing the feed per tooth, the values of the averaged diagonal length (L) decrease (from 13.48 to 5.87). For all analyzed cases, the highest values were achieved for GFRP milling and cooling with emulsion. The lowest values of this quantification were achieved for CFRP milling under dry machining conditions for each of the analyzed cases. An analysis of the longest diagonal length (L_max_) (Figure 4b) does not show a specific decreasing or increasing trend.

**Figure 4 materials-17-05981-f004:**
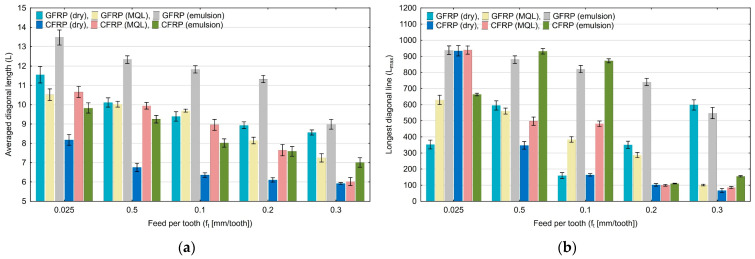
Influence of feed per tooth on recurrence quantifications: averaged diagonal length (**a**), longest diagonal line (**b**).

Figure 5a shows the longest vertical length (V_max_) results. An analysis of the diagram reveals no significant trend of the indicator change with increasing the feed per tooth. Figure 5b shows the values of the trapping time (TT) quantification. It is noticeable in this diagram that with increasing feed per tooth, the values of the trapping time decrease (from 4.98 to 2.92). In each analyzed group of feed per tooth, the highest values were achieved for milling GFRP without the use of coolants.

**Figure 5 materials-17-05981-f005:**
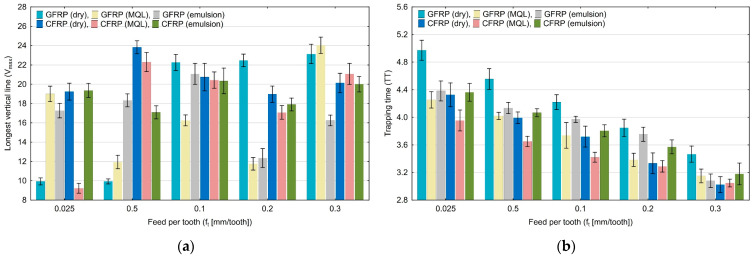
Influence of feed per tooth on recurrence quantifications: longest vertical line (**a**), trapping time (**b**).

The diagrams in Figure 6a,b show the values of the recurrence time of the first type (T_1_) and the recurrence time of the second type (T_2_). For the T_1_ quantification, there is no clear upward or downward trend. The marked standard deviations show that the index values have a large scatter. For the T_2_ quantification, there is a downward trend from 38.25 to 17.01 with an increase in the feed per tooth. It is noticeable that the highest values are obtained for the dry milling of GFRP.

**Figure 6 materials-17-05981-f006:**
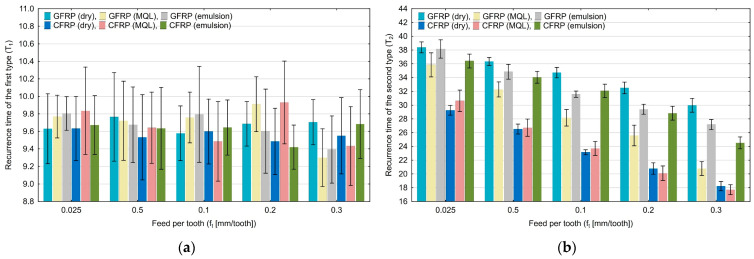
Influence of feed per tooth on recurrence quantifications: recurrence time of the first type (**a**), recurrence time of the second type (**b**).

The values of the recurrence period density entropy (RPDE) quantification presented in Figure 7a range from 0.79 to 0.48. There was no clear trend of change in this quantification with increasing the feed per tooth. Figure 7b presents the results of entropy (ENTR) as a function of the variable feed per tooth. It can be seen that by increasing the feed per tooth, the entropy values decrease from 3.20 to 2.02. The highest values of this quantification were achieved for the dry milling of GFRP.

**Figure 7 materials-17-05981-f007:**
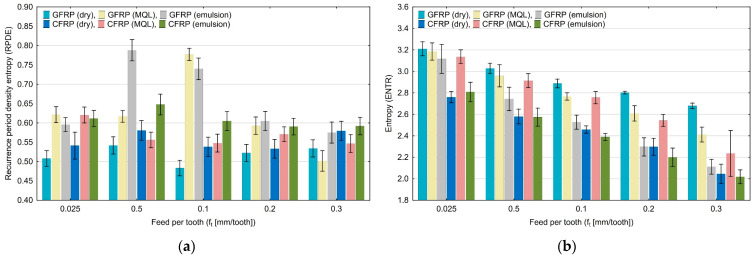
Influence of feed per tooth on recurrence quantifications: recurrence period density entropy (**a**), entropy (**b**).

Figure 8 shows the clustering coefficient values, but there is neither an increasing nor a decreasing trend in this diagram.

Based on the diagrams, it can be concluded that feed per tooth has a significant effect on recurrence quantifications. There are recurrence quantifications that show a decreasing or increasing trend depending on the type of tested polymer composite and the type of cooling environment. An increase in the feed per tooth value causes a decrease in the values of recurrence quantifications such as laminarity (LAM), averaged diagonal length (L), trapping time (TT), recurrence time of the second type (T_2_), and entropy (ENTR), for each of the tested combinations of polymer composite and cooling medium. Determinism (DET) is the only quantification that increases with feed per tooth for each composite material in “dry” machining and the two types of cooling liquids, i.e., MQL technology and emulsion cooling. Regarding other recurrence quantifications such as the longest diagonal line (L_max_), longest vertical line (V_max_), recurrence time of the first type (T_1_), recurrence period density entropy (RPDE), and clustering coefficient (CC), the decreasing or increasing trends only occurred for some types of cooling methods for individual polymer composites.

**Figure 8 materials-17-05981-f008:**
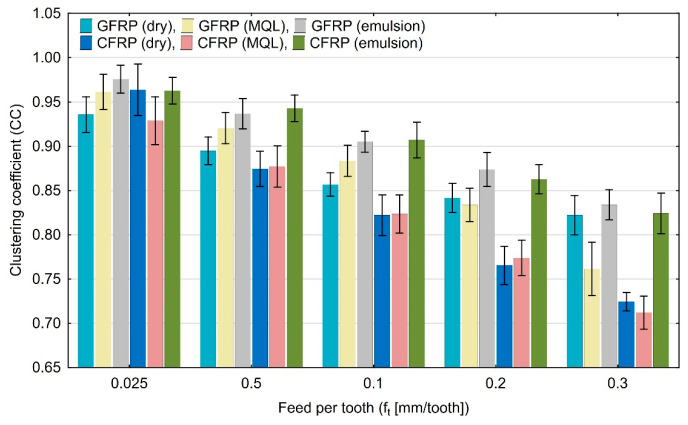
Influence of feed per tooth on recurrence quantifications: clustering coefficient.

The study on the normality of distribution confirmed that the recurrence quantification had a normal distribution. The results were used to perform an ANOVA analysis of variance. Table 2 and Table 3 show the ANOVA results for the tested recurrence quantifications. Table 2 shows the effect of feed per tooth on recurrence quantifications in GFRP milling and Table 3 lists the results for CFRP materials, for three cases of cooling.

The analysis shows that for most cases the feed per tooth has a significant impact on the obtained values of recurrence quantifications. For cases where the probability level *p* is lower than the assumed significance level (α = 0.05), and the value of the test statistic F_(4;10)_ is greater than the assumed F_α_ = 3.48 is denoted by “YES”. The results obtained from the ANOVA analysis prove that there are statistical differences in the mean values of the recurrence quantifications between the analyzed groups of variable feed per tooth.

The results obtained for the two different composite materials with variable feed per tooth are listed in Table 4. The “▬” sign in the table denotes the lack of a clear effect of the analyzed parameter on a given recurrence quantification, while “▲” and “▼” stand for the increase and decrease in a recurrence quantification value, respectively. In addition, the green color indicates cases that showed statistical differences in the tested quantifications. The red color indicates cases that did not show statistical differences in the quantifications.

### 3.3. Effect of Cutting Speed on Recurrence Quantifications

The study also determined the effect of cutting speed on recurrence quantifications in the milling of two polymer composites using cooling (MQL or emulsion) and under dry machining conditions.

Diagrams in Figure 9, Figure 10, Figure 11, Figure 12, Figure 13 and Figure 14 show the relationship between recurrence quantifications and variable cutting speed.

Figure 9a shows the effect of cutting speed on determinism (DET). A decrease in the DET values is noticeable with increased cutting speed. The values decrease from 0.94 to 0.53. The highest determinism values occurred for the dry milling of GFRP. The laminarity (LAM) results, plotted in Figure 9b, show a similar trend. With an increase in cutting speed, the values of this quantification decrease. The changes in this quantification range from 0.97 to 0.48.

**Figure 9 materials-17-05981-f009:**
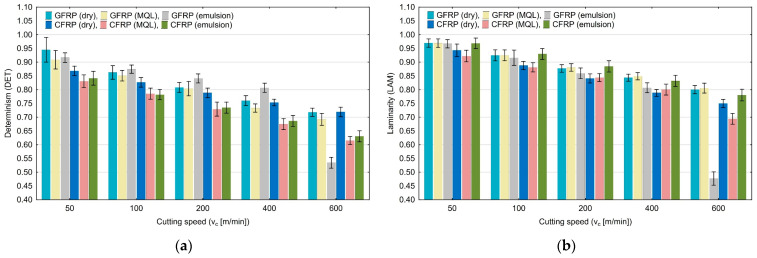
Influence of cutting speed on recurrence quantifications: determinism (**a**), laminarity (**b**).

The averaged diagonal length (L) quantification shown in Figure 10a increases from 7.30 to 24.23 within the tested range of variable cutting speed (50–600 m/min). The highest values were achieved for the highest tested cutting speed. The longest diagonal length (Lmax) quantification shown plotted in Figure 10b does not show a clear change with an increase in the cutting speed.

**Figure 10 materials-17-05981-f010:**
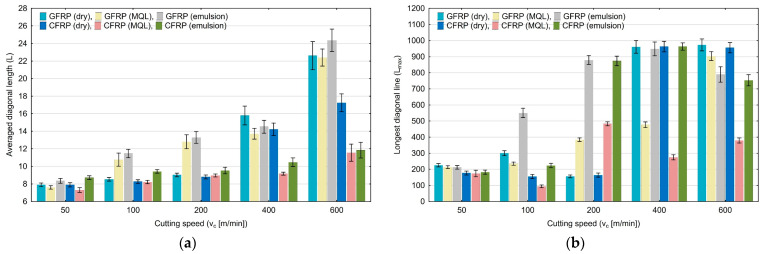
Influence of cutting speed on recurrence quantifications: averaged diagonal length (**a**), longest diagonal line (**b**).

Figure 11a,b show the results of the longest vertical line (V_max_) and trapping time (TT), the values of which decrease with the increase in cutting speed. The V_max_ quantification decreases from 68 to 4, and the TT quantification from 6.68 to 2.08. For both quantifications, the highest values were obtained for the cutting speed of 50 m/min.

The effect of cutting speed on the recurrence time of the first type (T_1_) and of the second type (T_2_) is plotted in Figure 12a and Figure 12b, respectively. For the first quantification, an increase in the cutting speed caused an increase in T_1_ from 8.36 to 10.80. The highest values were obtained for the cutting speed of 600 m/min. The T_2_ quantification values decreased from 48.65 to 13.38. The lowest value was obtained for GFRP milling with emulsion cooling.

For the recurrence period density entropy (RPDE) shown in Figure 13a, no upward or downward trend can be observed. The entropy (ENTR) results in Figure 13b show a downward trend with increasing cutting speed. The entropy values decrease from 3.16 to 1.86. The highest values were obtained for the cutting speed of 50 m/min.

**Figure 11 materials-17-05981-f011:**
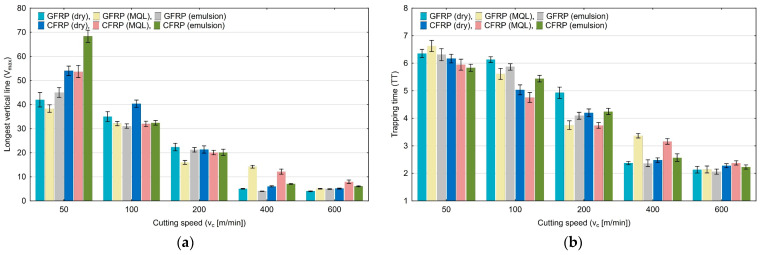
Influence of cutting speed on recurrence quantifications: longest vertical line (**a**), trapping time (**b**).

**Figure 12 materials-17-05981-f012:**
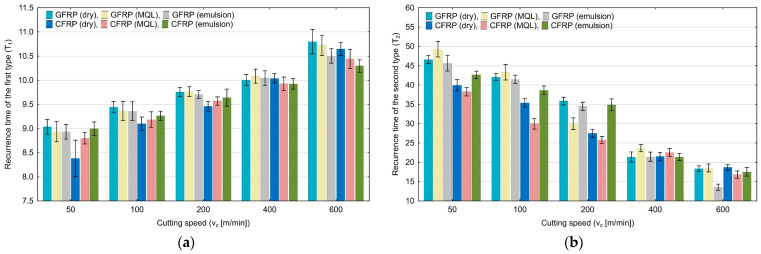
Influence of cutting speed on recurrence quantifications: recurrence time of the first type (**a**), recurrence time of the second type (**b**).

**Figure 13 materials-17-05981-f013:**
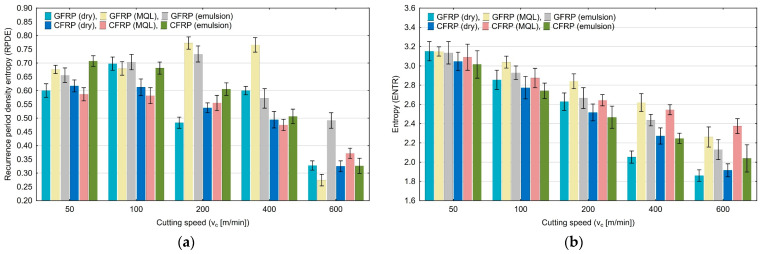
Influence of cutting speed on recurrence quantifications: recurrence period density entropy (**a**), entropy (**b**).

The clustering coefficient (CC) results plotted in Figure 14 do not show any significant change in the form of an increasing or decreasing trend with an increase in the cutting speed.

**Figure 14 materials-17-05981-f014:**
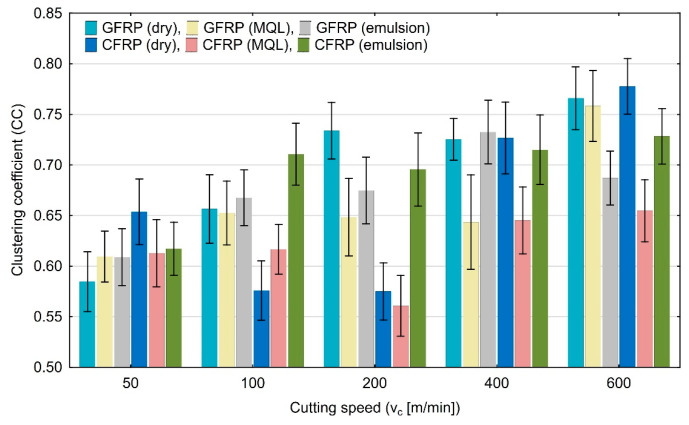
Influence of cutting speed on recurrence quantifications: clustering coefficient.

An analysis of the influence of the cutting speed allows us to state that there are recurrence quantifications that are sensitive to changes in this parameter. It can be observed that an increase in the cutting speed causes a decrease in the values of determinism (DET), laminarity (LAM), longest vertical line (V_max_), trapping time (TT), recurrence time of the second type (T_2_) and entropy (ENTR), for all tested polymer composites and for each of the tested cooling media. The cutting speed increase also causes an increase in the averaged diagonal length (L) and recurrence time of the first type (T_1_). Other recurrence quantifications increase or decrease their values only for certain composite materials and cooling conditions.

Table 5 and Table 6 list the ANOVA results of recurrence quantifications for variable cutting speed in the milling of GFRP and CFRP.

The ANOVA results demonstrate that the cutting speed has a significant impact on the obtained values of the recurrence quantifications in all tested cases. The results prove that there are statistical differences between the analyzed groups of variable cutting speed (denoted by green in Table 7).

Table 7 lists the results of the effect of variable cutting speed on the tested recurrence quantifications for polymer composites.

An analysis of the results showing the impact of variable technological parameters of milling on the recurrence quantifications reveals that there are change-sensitive recurrence quantifications that are common for all tested composite materials and applied cooling techniques. These quantifications include determinism (DET), laminarity (LAM), averaged diagonal length (L), trapping time (TT), recurrence time of the second type (T_2_), and entropy (ENTR).

## 4. Discussion

The application of recurrence methods in machining has shown the effectiveness of these techniques in this field. Nonlinear techniques have been successfully used in milling [36], drilling [38] and turning [39]. By analyzing signals in the form of cutting forces or vibrations, one can determine the phenomena and effects of machining. As materials that are made of at least two components, polymer composites are susceptible to defects, the detection of which is a priority if they are to be used to build responsible machine elements. This observation allowed us to formulate hypotheses about the possibility of detecting defects in composite milling by means of recurrence methods. The study involved an analysis of recurrence quantifications in order to detect the size and location of defects [29]. The results of the study allowed us to determine the size and depth of an artificial defect in relation to the tool diameter, so that the defect could be detectable. Positive results of artificial defect tests were then used to detect real defects inside the composite material using recurrence methods [51]. Their presence was confirmed by non-destructive methods first and then confirmed using recurrence methods by determining indicators sensitive to detecting such defects. Recurrence analyses were also used to determine the sensitivity of quantifications to the type of tool and material being machined [37]. The quantifications included determinism (DET), averaged diagonal length (L) and entropy (ENTR).

With previous studies in mind, this study also undertook to determine recurrence quantifications that were sensitive to the type of cooling used in the machining of the two different types of polymer composites. It was shown that there the quantifications such as determinism, laminarity, averaged diagonal length, trapping time, recurrence time of the second type, and entropy met the above hypothesis.

## 5. Conclusions

This paper presented the results of a study on the machining of glass and carbon fiber-reinforced plastics by milling. The milling process was carried out using variable technological parameters such as the feed per tooth ranging from 0.025 to0.3 mm/rev and the cutting speed ranging from 50 to60 m/min. In addition, the milling process was conducted with different cooling methods such as MQL technology and emulsion cooling. For comparison, the milling process was also carried out without the use of cooling liquids, i.e., dry. The results of the study demonstrate:Recurrence quantifications such as determinism (DET), laminarity (LAM), averaged diagonal length (L), trapping time (TT), recurrence time of the second type (T_2_) and entropy (ENTR) are suitable for determining the effect of feed per tooth on the two tested composite materials under three different milling conditions. The quantifications change in the following ranges: from 0.63 to 1.02 (DET), from 0.97 to 0.69 (LAM), from 13.48 to 5.87 (L), from 4.98 to 2.92 (TT), 38.25 to 17.01 (T_2_), from 3.20 to 2.02 (ENTR).For variable cutting speed, suitable for determining the effect this parameter are: determinism (DET), laminarity (LAM), averaged diagonal length (L), longest vertical line (V_max_), trapping time (TT), recurrence time of the first type (T_1_), recurrence time of the second type (T_2_), and entropy (ENTR). The quantifications change in the following ranges: from 0.94 to 0.53 (DET), from 0.97 to 0.48 (LAM), from 7.30 to 24.23 (L), from 68 to 4 (V_max_), from 6.68 to 2.08 (TT), from 8.36 to 10.80 (T_1_), from 48.65 to 13.38 (T_2_), from 3.16 to 1.86 (ENTR).The results also make it possible to determine the recurrence quantifications that are common for both tested materials, for three different cooling conditions and two variable parameters of milling. The quantifications that are sensitive to changes in technological parameters of milling for glass and carbon fiber-reinforced plastics with different cooling include determinism (DET), laminarity (LAM), averaged diagonal length (L), trapping time (TT), recurrence time of the second type (T_2_), and entropy (ENTR). It is possible to determine common ranges of changes in these quantifications for both tested variables of milling technological parameters: 0.63–0.94 (DET), 0.69–0.97 (LAM), 7.30–13.48 (L), 2.92–4.98 (TT), 17.01–38.25 (T_2_), 2.02–3.16 (ENTR).Statistical analysis showed that the variables tested, i.e., feed per tooth and cutting speed, have a normal distribution. On this basis, ANOVA analysis was performed, which showed that there are statistical differences in the mean values of recurrence quantifications for the two tested composite materials in three cooling cases between the analyzed groups of variables (feed per tooth and cutting speed).

Based on the current article and previous works, it can be stated that it is possible to predict the type of material, tool, and type of cooling using recurrence quantifications. In order to extrapolate the influence of variable factors, research is planned to determine recurrence quantifications that clearly indicate the type of material, the tool used, and the type of cooling with variable technological parameters.

Future research directions should also focus on determining recurrence quantifications that are sensitive to cutting tool wear. In addition, studies should be conducted on other types of cutting tools in order to determine universal recurrence quantifications that could detect the type of cooling and polymer composite being tested.

## Figures and Tables

**Figure 1 materials-17-05981-f001:**
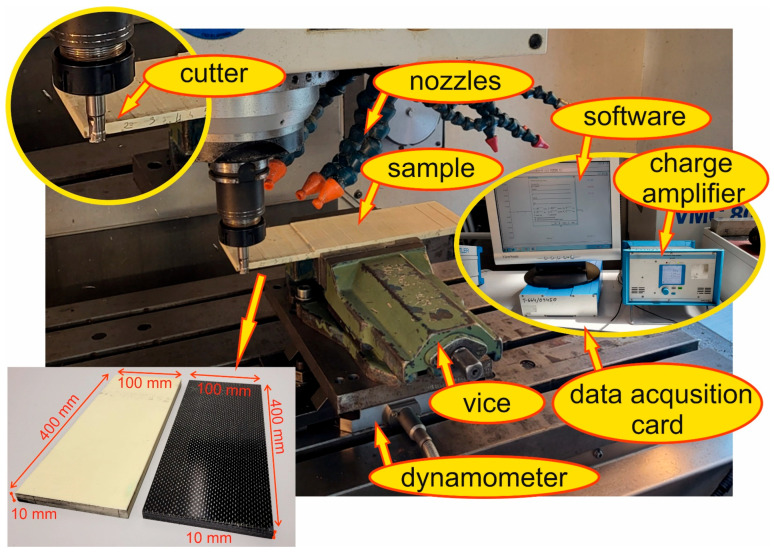
Scheme of the research methodology.

**Figure 2 materials-17-05981-f002:**
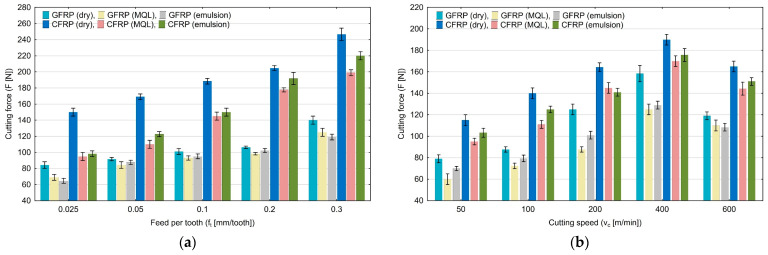
Influence of feed per tooth (**a**) and cutting speed (**b**) on the maximum cutting force.

**Table 1 materials-17-05981-t001:** Recurrence quantifications with their abbreviations and formulas [46,47,48,49,50].

Recurrence Quantification	Abbreviations	Formula
determinism	DET	$DET=\frac{\sum_{l=l_{\min}}^{N}lP(l)}{\sum_{i,j=1}^{N}R_{i,j}}$
laminarity	LAM	$LAM=\frac{\sum_{v=v_{\min}}^{N}vP(v)}{\sum_{v=1}^{N}vP(v)}$
averaged diagonal length	L	$L=\frac{\sum_{l=l_{\min}}^{N}lP(l)}{\sum_{l=l_{\min}}^{N}P(l)}$
longest diagonal line	L_max_	$L_{\max}=\max(\{l_{i};i=1\ldots N_{l}\})$
longest vertical line	V_max_	$V_{\max}=\max(\{v_{i};i=1\ldots N_{v}\})$
trapping time	TT	$TT=\frac{\sum_{v=v_{\min}}^{N}vP(v)}{\sum_{v=v_{\min}}^{N}P(v)}$
recurrence time of the first type	T1	$T_{1}=\left|\left\{\left\{i,j:{\bar{x}}_{i},{\bar{x}}_{j}\right\}\in R_{i}\right\}\right|$
recurrence time of the second type	T2	$T_{2}=\left|\left\{\left\{i,j:{\bar{x}}_{i},{\bar{x}}_{j}\right\}\in R_{i}\right\}{\bar{x}}_{j}\notin R_{i}\right|$
recurrence period density entropy	RPDE	$\mathrm{RPDE}=-\frac{1}{\ln v_{MAX}}\sum_{v=1}^{v_{MAX}}H_{v}(v)\ln H_{v}(v)$
entropy	ENTR	$ENTR=-\sum_{l=l_{\min}}^{N}p(l)\ln p(l)$
clustering coefficient	CC	$CC=\sum_{i=1}^{N}\frac{\sum_{j,k=1}^{N}R_{i,j}^{m,\varepsilon }R_{j,k}^{m,\varepsilon }R_{k,i}^{m,\varepsilon }}{\sum_{j=1}^{N}R_{i,j}^{m,\varepsilon }}$

**Table 2 materials-17-05981-t002:** ANOVA analysis of variance results of recurrence quantifications in GFRP milling conducted with variable feed per tooth and constant cutting speed.

	f_t_									
		GFRP (Dry)			GFRP (MQL)			GFRP (Emulsion)		
	DF	F	*p*	Significance	F	*p*	Significance	F	*p*	Significance
DET	4	34.4948	0.0000	YES	36.8772	0.0000	YES	36.3657	0.0000	YES
LAM	4	19.4945	0.0000	YES	39.7627	0.0000	YES	33.8616	0.0000	YES
L	4	60.0868	0.0000	YES	144.9941	0.0000	YES	128.7384	0.0000	YES
L_max_	4	154.0493	0.0000	YES	395.9196	0.0000	YES	99.4914	0.0000	YES
V_max_	4	310.4038	0.0000	YES	153.4973	0.0000	YES	43.9394	0.0000	YES
TT	4	61.3393	0.0000	YES	43.1010	0.0000	YES	74.5517	0.0000	YES
T_1_	4	0.1202	0.9721	NO	1.4392	0.2910	NO	0.4747	0.7538	NO
T_2_	4	49.8658	0.0000	YES	57.2696	0.0000	YES	67.5923	0.0000	YES
RPDE	4	3.4131	0.0000	YES	70.4285	0.0000	YES	43.4457	0.0000	YES
ENTR	4	69.1277	0.0000	YES	47.9796	0.0000	YES	49.7221	0.0000	YES
CC	4	6.3113	0.0000	YES	15.7022	0.0000	YES	18.8278	0.0000	YES

**Table 3 materials-17-05981-t003:** ANOVA results of recurrence quantifications in CFRP milling conducted with variable feed per tooth and constant cutting speed.

	f_t_									
		CFRP (Dry)			CFRP (MQL)			CFRP (Emulsion)		
	DF	F	*p*	Significance	F	*p*	Significance	F	*p*	Significance
DET	4	107.3677	0.0000	YES	38.2581	0.0000	YES	56.8993	0.0000	YES
LAM	4	56.2555	0.0000	YES	43.2718	0.0000	YES	29.257	0.0000	YES
L	4	78.5238	0.0000	YES	159.4220	0.0000	YES	75.0226	0.0000	YES
L_max_	4	1040.5038	0.0000	YES	1132.0394	0.0000	YES	4207.6445	0.0000	YES
V_max_	4	11.4602	0.0000	YES	115.4405	0.0000	YES	7.5817	0.0000	YES
TT	4	42.1016	0.0000	YES	41.6120	0.0000	YES	49.3826	0.0000	YES
T_1_	4	0.0584	0.9926	NO	0.6701	0.6274	NO	0.2734	0.8885	NO
T_2_	4	125.9915	0.0000	YES	59.4650	0.0000	YES	73.2435	0.0000	YES
RPDE	4	2.2572	0.1352	NO	6.2764	0.0000	YES	2.9637	0.0745	NO
ENTR	4	48.1846	0.0000	YES	29.8602	0.0000	YES	51.5395	0.0000	YES
CC	4	11.8809	0.0000	YES	74.3349	0.0000	YES	17.0425	0.0000	YES

**Table 4 materials-17-05981-t004:** Effect of feed per tooth on recurrence quantifications.

Recurrence Quantification	GFRP (Dry)	GFRP (MQL)	GFRP (Emulsion)	CFRP (Dry)	CFRP (MQL)	CFRP (Emulsion)
DET	▲	▲	▲	▲	▲	▲
LAM	▼	▼	▼	▼	▼	▼
L	▼	▼	▼	▼	▼	▼
L_max_	▬	▼	▼	▼	▼	▬
V_max_	▲	▬	▬	▬	▬	▬
TT	▼	▼	▼	▼	▼	▼
T_1_	▬	▬	▬	▬	▬	▬
T_2_	▼	▼	▼	▼	▼	▼
RPDE	▬	▬	▬	▬	▬	▬
ENTR	▼	▼	▼	▼	▼	▼
CC	▬	▼	▼	▼	▼	▬

**Table 5 materials-17-05981-t005:** ANOVA analysis results of recurrence quantifications in GFRP milling conducted with variable cutting speed and constant feed per tooth.

	v_c_									
		GFRP (Dry)			GFRP (MQL)			GFRP (Emulsion)		
	DF	F	*p*	Significance	F	*p*	Significance	F	*p*	Significance
DET	4	33.8856	0.0000	YES	39.3256	0.0000	YES	233.9537	0.0000	YES
LAM	4	55.1335	0.0000	YES	46.6379	0.0000	YES	245.7661	0.0000	YES
L	4	159.1149	0.0000	YES	180.6811	0.0000	YES	159.0247	0.0000	YES
L_max_	4	747.6760	0.0000	YES	883.0243	0.0000	YES	231.0050	0.0000	YES
V_max_	4	287.8355	0.0000	YES	684.8462	0.0000	YES	807.2872	0.0000	YES
TT	4	646.6176	0.0000	YES	374.6439	0.0000	YES	567.6526	0.0000	YES
T_1_	4	53.5879	0.0000	YES	44.2957	0.0000	YES	48.6969	0.0000	YES
T_2_	4	459.2659	0.0000	YES	205.8044	0.0000	YES	329.0053	0.0000	YES
RPDE	4	143.1085	0.0000	YES	257.8451	0.0000	YES	33.8428	0.0000	YES
ENTR	4	123.3312	0.0000	YES	59.8577	0.0000	YES	53.1961	0.0000	YES
CC	4	18.8785	0.0000	YES	7.3130	0.0000	YES	6.7978	0.0000	YES

**Table 6 materials-17-05981-t006:** ANOVA analysis results of recurrence quantifications in CFRP milling conducted with variable cutting speed and constant feed per tooth.

	v_c_									
		CFRP (Dry)			CFRP (MQL)			CFRP (Emulsion)		
	DF	F	*p*	Significance	F	*p*	Significance	F	*p*	Significance
DET	4	38.0839	0.0000	YES	46.1284	0.0000	YES	49.9449	0.0000	YES
LAM	4	63.5941	0.0000	YES	67.7810	0.0000	YES	41.6957	0.0000	YES
L	4	34.3818	0.0000	YES	34.3818	0.0000	YES	17.1526	0.0000	YES
L_max_	4	1134.2785	0.0000	YES	327.8653	0.0000	YES	692.8863	0.0000	YES
V_max_	4	790.8896	0.0000	YES	511.6599	0.0000	YES	1084.0006	0.0000	YES
TT	4	474.9553	0.0000	YES	294.4479	0.0000	YES	582.4010	0.0000	YES
T_1_	4	56.6394	0.0000	YES	57.3806	0.0000	YES	45.6489	0.0000	YES
T_2_	4	201.8352	0.0000	YES	166.5221	0.0000	YES	280.5646	0.0000	YES
RPDE	4	71.8980	0.0000	YES	42.5829	0.0000	YES	129.3214	0.0000	YES
ENTR	4	69.2718	0.0000	YES	29.1959	0.0000	YES	35.8832	0.0000	YES
CC	4	25.8142	0.0000	YES	4.3809	0.0000	YES	6.0018	0.0000	YES

**Table 7 materials-17-05981-t007:** Effect of cutting speed on recurrence quantifications.

Recurrence Quantification	GFRP (Dry)	GFRP (MQL)	GFRP (Emulsion)	CFRP (Dry)	CFRP (MQL)	CFRP (Emulsion)
DET	▼	▼	▼	▼	▼	▼
LAM	▼	▼	▼	▼	▼	▼
L	▲	▲	▲	▲	▲	▲
L_max_	▬	▲	▬	▬	▬	▬
V_max_	▼	▼	▼	▼	▼	▼
TT	▼	▼	▼	▼	▼	▼
T_1_	▲	▲	▲	▲	▲	▲
T_2_	▼	▼	▼	▼	▼	▼
RPDE	▬	▬	▬	▼	▼	▼
ENTR	▼	▼	▼	▼	▼	▼
CC	▬	▬	▬	▬	▬	▲

## Data Availability

The original contributions presented in this study are included in the article. Further inquiries can be directed to the corresponding author.

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
