# Peer review of "Influence of the Cooling Method on Cutting Force and Recurrence Analysis in Polymer Composite Milling"

_materials, 2024, doi:10.3390/ma17235981_

Round 1

Reviewer 1 Report

Comments and Suggestions for Authors

please see the attached file

Author Response

I would like to thank the Reviewer for the time spent on carefully reviewing this work and for their valuable deep insight and comments. I feel that this paper is now clearer, more thoroughly discussed and better-referenced. The work has been revised to address the reviewer suggestion. Please find hereafter a point-by-point reply to the comments and suggestions. Red words in article indicate changes (“track changes” option) from the original text of the manuscript. 

General comment from Reviewer:  The manuscript is interesting. Introduction section is well written and comprehensive, contains many relevant references, and gives the reader a very good overall insight into the subject. The other sections are also well written. However, few specific comments are present to improve the quality of the manuscript.

Response:

I would like to thank the Reviewer for his opinion. We feel to be obligated to answer for points mention in the review. The paper has been modified and improved. We believe that now is clearer.

Comments to the title:  I propose to adjust the „feed force“ to „cutting force“ which is commonly used.

Response:

Thank you for these suggestions. The name of force in title was changed for "cutting force".

Comments to the abstract:  I propose to adjust the „feed force“ to „cutting force“ which is commonly used, „feed per revolution“ is not the common parameter used when milling. It is mainly used when turning. I propose to use peed per tooth in this case.

Response:

Thank you for these suggestions. In abstract “feed force” was changed on “cutting force”. Of course “feed per revolution” was also changed for “feed per tooth”.

General comments to each sections:  Through the whole manuscript it is necessary to adjust:

- I propose to adjust the „feed force“ to „cutting force“ which is commonly used in the whole manuscript,

-„feed per revolution“ is not the common parameter used when milling. It is mainly used when turning. I propose to use peed per tooth in the whole manuscript. This will surely help to increase the results presentation quality.

- Please check all the qraphs. There units are missing in lot of cases of the vertical axes.

Response:

Thank you for these suggestions. In manuscript “feed force” was changed on “cutting force” in text and in graphs. Moreover “feed per revolution” was changed for “feed per tooth” in text of manuscript and in all graphs. The vertical axes of the graphs show the values ​​of recurrence quantifications. These are dimensionless quantifications, so they have no units.

Comments to Section 1:  Introduction section is very well written and comprehensive, contains many relevant references, and gives the reader a very good overall insight into the subject.

- Nevertheless, I also recommend to include the following important papers. Within the introduction section, please also include at least the following two sources of the state of the art on this issue: https://doi.org/10.1016/j.jmatprotec.2004.07.146 and https://doi.org/10.1016/j.rineng.2022.100762 These papers show how important it is to use feed rate control especially in the case of curved toolpaths, where the angular velocity of movement is involved, which happens also in standard applications of composite milling. It deals with achieving of the constant feed per tooth in the contact point between the tool and the workpiece. This is related to your manuscript as you are dealing with the feed rate (or feed per revolution or feed per tooth).

Response:

I would like to thank for opinion and recommendation. The papers were added to the text of the article and placed in the references. These are interesting works that will certainly enrich the literature review of this article.

I appreciate for Reviewer warm work earnestly, and hope that the corrections will meet with approval. Once again, I thank you very much for your comment and suggestion.

Yours sincerely,

Krzysztof Ciecieląg

Reviewer 2 Report

Comments and Suggestions for Authors

The manuscript appears to be clear and well-structured, with each section logically progressing from the introduction of the problem to the experimental design, results, and discussion. The topic is highly relevant to the field of materials science and manufacturing, particularly in the context of improving machining processes for polymer composites, which are widely used in industries such as aerospace and automotive due to their high strength-to-weight ratios.

The manuscript includes a substantial number of references, and while some are recent, others are over five years old. There are several recent citations within the last five years, especially concerning sustainable cooling methods and machining of composites. There does not appear to be an excessive number of self-citations.

The manuscript presents an experimental design to test the stated hypotheses. The study uses controlled and varied parameters, including feed rate, cutting speed, and different cooling methods, to isolate the effects of these variables on the milling of GFRP and CFRP composites.

Based on the details provided in the methods section, the study's results should be reproducible. The manuscript describes the experimental setup, including the equipment used (such as the specific machining centre, cutter type, and dynamometer), the exact dimensions of the test samples, and the software employed for data analysis. Parameters like feed rate, cutting speed, and cooling method are specified, which should enable other researchers to replicate the study.

The figures, tables presented in the manuscript are good. They effectively illustrate the impact of different cooling methods on feed force. Recurrence quantifications like determinism (DET), laminarity (LAM), and entropy (ENTR) are summarized well in tables, allowing readers to quickly assess which quantifications are sensitive to changes in machining parameters.

The data presentation is clear, and the statistical interpretations are consistent throughout the manuscript. Although no extensive statistical analysis is reported, the trends in recurrence quantifications are clearly described in the results section. However, if further analysis were to be added, including statistical significance testing of the observed trends (such as ANOVA for recurrence quantifications across cooling methods), it could strengthen the reliability of the conclusions.

The conclusions appropriately highlight the relevance of the reported quantifications as tools for assessing machining performance in polymer composites.

The ethics and data availability statements in the manuscript are minimal but appear adequate for this type of study.

 Author Response

I would like to thank the Reviewer for the time spent on carefully reviewing this work and for their valuable deep insight and comments. I feel that this paper is now clearer, more thoroughly discussed and better-referenced. The work has been revised to address the reviewer suggestion. Red words in article indicate changes (“track changes” option) from the original text of the manuscript. 

General comment from Reviewer:  The manuscript appears to be clear and well-structured, with each section logically progressing from the introduction of the problem to the experimental design, results, and discussion. The topic is highly relevant to the field of materials science and manufacturing, particularly in the context of improving machining processes for polymer composites, which are widely used in industries such as aerospace and automotive due to their high strength-to-weight ratios.The manuscript includes a substantial number of references, and while some are recent, others are over five years old. There are several recent citations within the last five years, especially concerning sustainable cooling methods and machining of composites. There does not appear to be an excessive number of self-citations.The manuscript presents an experimental design to test the stated hypotheses. The study uses controlled and varied parameters, including feed rate, cutting speed, and different cooling methods, to isolate the effects of these variables on the milling of GFRP and CFRP composites.Based on the details provided in the methods section, the study's results should be reproducible. The manuscript describes the experimental setup, including the equipment used (such as the specific machining centre, cutter type, and dynamometer), the exact dimensions of the test samples, and the software employed for data analysis. Parameters like feed rate, cutting speed, and cooling method are specified, which should enable other researchers to replicate the study.The figures, tables presented in the manuscript are good. They effectively illustrate the impact of different cooling methods on feed force. Recurrence quantifications like determinism (DET), laminarity (LAM), and entropy (ENTR) are summarized well in tables, allowing readers to quickly assess which quantifications are sensitive to changes in machining parameters.The data presentation is clear, and the statistical interpretations are consistent throughout the manuscript. Although no extensive statistical analysis is reported, the trends in recurrence quantifications are clearly described in the results section. However, if further analysis were to be added, including statistical significance testing of the observed trends (such as ANOVA for recurrence quantifications across cooling methods), it could strengthen the reliability of the conclusions.The conclusions appropriately highlight the relevance of the reported quantifications as tools for assessing machining performance in polymer composites. The ethics and data availability statements in the manuscript are minimal but appear adequate for this type of study.

Response:

I would like to thank the Reviewer for his opinion. In the opinion there is an important suggestion to add ANOVA statistical analysis. ANOVA analysis was performed and added to the article content. The analysis showed that there are statistical differences in the mean values ​​of recurrence quantifications for the two tested composite materials in three cooling cases between the analyzed groups of variables (feed per tooth and cutting speed). This analysis was added in Chapter 3: Results.

I appreciate for Reviewer warm work earnestly, and hope that the corrections will meet with approval. Once again, I thank you very much for your comment and suggestion.

In the attachment also is this Response.

Yours sincerely,

Krzysztof Ciecieląg

Reviewer 3 Report

Comments and Suggestions for Authors

The study addresses a relevant topic within manufacturing engineering and the machining of composite materials, combining experimental analysis with nonlinear tools, such as recurrence metrics. This approach is innovative and has the potential to improve the understanding of dynamic phenomena in machining processes. However, in my opinion, there are critical areas that require improvement to enhance the impact and applicability of the research.

  • Although the study identifies correlations between technological parameters, cooling methods, and recurrence metrics, it does not develop any mathematical or predictive models that could anticipate behavior under different conditions. The author should justify this significant shortcoming.
  • Recurrence metrics are not compared with other traditional methodologies (e.g., spectral or statistical analysis), which makes it difficult to assess whether these metrics offer a significant advantage. The author should address this comparison.
  • The conclusions are limited to the specific experimental conditions, without sufficient discussion on how to extrapolate the results to other materials, tools, or cooling methods.
  • The figures, while useful, suffer from visual overload in graphs with multiple parameters. The author could combine tables with graphs and reduce the number of figures. Choosing more reader-friendly colors and clearer, more visible graphics would improve the document’s appearance.

Author Response

I would like to thank the Reviewer for the time spent on carefully reviewing this work and for their valuable deep insight and comments. I feel that this paper is now clearer, more thoroughly discussed and better-referenced. The work has been revised to address the reviewer suggestion. Please find hereafter a point-by-point reply to the comments and suggestions. Red words in article indicate changes (“track changes” option) from the original text of the manuscript. 

General comment from Reviewer:  The study addresses a relevant topic within manufacturing engineering and the machining of composite materials, combining experimental analysis with nonlinear tools, such as recurrence metrics. This approach is innovative and has the potential to improve the understanding of dynamic phenomena in machining processes. However, in my opinion, there are critical areas that require improvement to enhance the impact and applicability of the research.

Response:

I would like to thank the Reviewer for his opinion. We feel to be obligated to answer for points mention in the review. The paper has been modified and improved. We believe that now is clearer.

Comment 1:  Although the study identifies correlations between technological parameters, cooling methods, and recurrence metrics, it does not develop any mathematical or predictive models that could anticipate behavior under different conditions. The author should justify this significant shortcoming.

Response:

Thank you for this question. Recurrence quantifications are calculated using mathematical formulas using Matlab software. In next work in future, it is planned to derive a formula that will allow predicting the values ​​of quantifications based on variable technological parameters of milling.

 Comment 2:  Recurrence metrics are not compared with other traditional methodologies (e.g., spectral or statistical analysis), which makes it difficult to assess whether these metrics offer a significant advantage. The author should address this comparison.

Response:

Thank you for these suggestions. In the article was added ANOVA analysis, which showed that the variables tested, i.e. feed per tooth and cutting speed, have a normal distribution. Additionally, ANOVA analysis was also performed and added to the article content. The analysis showed that there are statistical differences in the mean values of recurrence quantifications for the two tested composite materials in three cooling cases between the analyzed groups of variables (feed per tooth and cutting speed). This analysis was added in Chapter 3: Results. Standard deviations have also been added to the graphs.

Comment 3:  The conclusions are limited to the specific experimental conditions, without sufficient discussion on how to extrapolate the results to other materials, tools, or cooling methods.

Response:

Thank you for these suggestions. In previous works, the influence of the type of material and tool on recurrence quantifications during milling and drilling was analyzed. In the work on the influence of the type of material and tools on recurrence quantifications during milling, it was shown that the most sensitive quantifications are determinism, average diagonal length and entropy. It was shown that the indices change with the change of the type of material and tool, showing decreasing or increasing trends. The tendency was related to the quality of the tool for machining polymer composites. In the work on drilling polymer composites, it was shown that there are recurrence quantifications that change (increasing or decreasing) with the change of technological parameters of drilling. The quantifications sensitive to the drilling process include determinism, longest diagonal line, entropy and laminarity.

In the conclusions was added a discussion on extrapolating the results to other materials, tools, or cooling methods. In order to extrapolate the influence of the material, tool and cooling method on the recurrence quantifications, research is planned that takes into account all factors and derives quantifications that clearly indicate the conditions of the analysis. The current research has shown the sensitivity of recurrence methods to machining, so it can be hypothesized that there are values ​​of recurrence quantifications that clearly determine the input and output factors of the cutting process such as materials, tools, or cooling methods.

Comment 4:  The figures, while useful, suffer from visual overload in graphs with multiple parameters. The author could combine tables with graphs and reduce the number of figures. Choosing more reader-friendly colors and clearer, more visible graphics would improve the document’s appearance.

Response:

I would like to thank for opinion and suggestions. In order to improve the graphs, the colors were changed to more friendly ones. The bars in the graphs were also widened and the distances between the bars were reduced. Standard deviations were also added to the graphs. To reduce visual overload, the tables of the influence of the milling technological parameters (feed per tooth and cutting speed) on recurrence quantifications were placed directly behind the graphs. The tables also added colors showing the statistical significance of the parameter. Green color means that the analyzed parameter is significant for the recurrence quantification. Figures 3 and 4 have been divided into pairs. Descriptions have been added under each pair of graphs. This is to provide a detailed discussion of the influence of milling technological parameters on recurrence quantifications.

I appreciate for Reviewer warm work earnestly, and hope that the corrections will meet with approval. Once again, I thank you very much for your comment and suggestion.

In the attachment also is this Response.

Yours sincerely,

Krzysztof Ciecieląg

Reviewer 4 Report

Comments and Suggestions for Authors

This paper is stated on the "Influence of cooling method on feed force and recurrence analysis in polymer composite milling"

(1) According to the ithenticate report, the percent match of this paper with the other papers is too high about 34%.

(2) The abstract section should be written again. What is this main method used in this study. Add the numerical results to this section.

(3) The feed force is the moving force during the milling process?

(4) Why the authors chose the "polymer composite milling"? The other materials also can be used in this milling process.

(5) The introduction section is too long, the tables and formula must be removed from this section.

(6) The image and dimension of material should be added.

(7) The quality of Fig. 1 is too low? Modify the image of Fig. 1.

(8) Too many images in Fig. 4. More explanation of each image should be showed.

(9) The conclusions is not good. Used (1), (2), (3),.....for this section.

Comments on the Quality of English Language

The English could be improved to more clearly express the research.

Author Response

I would like to thank the Reviewer for the time spent on carefully reviewing this work and for their valuable deep insight and comments. I feel that this paper is now clearer, more thoroughly discussed and better-referenced. The work has been revised to address the reviewer suggestion. Please find hereafter a point-by-point reply to the comments and suggestions. Red words in article indicate changes (“track changes” option) from the original text of the manuscript. 

Comment 1:  According to the ithenticate report, the percent match of this paper with the other papers is too high about 34%.

Response:

I would like to thank for this comment. In the introduction to the article, some sentences were removed. Table 1 was moved to the chapter related to methodology, and descriptions of quantifications that can be found in other works were removed. Some content was rewritten in the chapter related to methodology. In the chapter related to research results, detailed descriptions of each of the graphs were added. In addition, figures were separated to improve clarity. The research results were subjected to statistical analysis ANOVA, and the results were placed and described in the body of the article. Standard deviations were also added to the graphs. In the conclusion, points presenting conclusions were added. Directions of the future research were also added.

Comment 2:  The abstract section should be written again. What is this main method used in this study. Add the numerical results to this section.

Response:

Thank you for these suggestions. The abstract has been improved. The novelty of the work has been clearly indicated. It has also been described what research method was used in the article. The abstract has also added numerical results resulting from the conducted research.

Comment 3:  The feed force is the moving force during the milling process?

Response:

Thank you for these question. The cutting force was a factor analyzed during milling, as well as a signal for analysis using recurrence methods. In the methods, the force signal was reconstructed using the delay method. The time delay d, the embedding dimension m and the threshold ε were selected and on this basis it was possible to calculate the quantifications and prepare the graphs. The direction of the feed force was consistent with the direction of movement of the cutter.

Comment 4:  Why the authors chose the "polymer composite milling"? The other materials also can be used in this milling process.

Response:

I would like to thank for question. The author analyzes composites because it is a subject related to his scientific activity. The study of the cutting process of composites is still an inexhaustible subject. Of course, studies using recurrence quantifications of other materials are planned. Studies of aluminum alloys and titanium alloys are planned in order to detect defects, burrs and scratches on their surfaces during milling.

Comment 5:  The introduction section is too long, the tables and formula must be removed from this section.

Response:

I would like to thank for suggestions. The introduction has been shortened by sentences presenting general knowledge. Table 1 has been moved to the chapter related to methodology. The description of symbols used in the formulas for recurrence quantifications has been moved along with the table.

Comment 6:  The image and dimension of material should be added.

Response:

I would like to thank for suggestions. The image of the samples with dimensions has been added to Figure 1. The text also includes information about the dimensions and method of prepare the samples: “The composite samples were in the form of plates with the dimensions of 10x100x400 (Fig. 1). The 10 mm thick sample was formed by layers of 40 prepregs in a 0°-90° arrangement. The composite samples were fabricated in an autoclave where they were heated for 2 h at 177°C (+/-2°C) at a pressure of 0.3 MPa. Prior to putting them into the autoclave, the samples were prepared in a special room where cleanliness was maintained and the number of solid particles per 1 m3 did not exceed 10,000. The temperature in the room was maintained in the range of 18°C-30°C and the humidity level was below 60%.”

Comment 7:  The quality of Fig. 1 is too low? Modify the image of Fig. 1.

Response:

I would like to thank for opinion and suggestions. Figure 1 has been improved in terms of quality and a sample photo with dimensions has been added. The figure is inserted in 600 dpi resolution. The built-in pdf creation system may reduce the quality of the figures. If the review process is completed, the figure will be sent to editors in high quality.

Comment 8:  Too many images in Fig. 4. More explanation of each image should be showed..

Response:

I would like to thank for opinion and suggestions. Figures 3 and 4 have been divided into pairs. Descriptions have been added under each pair of graphs. This is to provide a detailed discussion of the influence of milling technological parameters on recurrence quantifications.

To reduce visual overload, the tables of the influence of the milling technological parameters (feed per tooth and cutting speed) on recurrence quantifications were placed directly behind the graphs. The tables also added colors showing the statistical significance of the parameter. Green color means that the analyzed parameter is significant for the recurrence quantification.

Comment 9:  The conclusions is not good. Used (1), (2), (3),.....for this section..

Response:

I would like to thank for suggestions. The conclusions were presented in points. This way of presenting the conclusions certainly improves readability and allows you to familiarize yourself with the most important results.

I appreciate for Reviewer warm work earnestly, and hope that the corrections will meet with approval. Once again, I thank you very much for your comment and suggestion.

In the attachment also is this Response.

Yours sincerely,

Krzysztof Ciecieląg

Round 2

Reviewer 3 Report

Comments and Suggestions for Authors

The author has taken the trouble to respond adequately to most of the issues raised in a satisfactory manner.  I thank him for his efforts. 

Reviewer 4 Report

Comments and Suggestions for Authors

The quality of paper has been improved.

It can be accepted.

Comments on the Quality of English Language

The English could be improved to more clearly express the research.